# Sjögren’s Syndrome-Related Organs Fibrosis: Hypotheses and Realities

**DOI:** 10.3390/jcm11123551

**Published:** 2022-06-20

**Authors:** Margherita Sisto, Domenico Ribatti, Sabrina Lisi

**Affiliations:** Department of Basic Medical Sciences, Neurosciences and Sensory Organs (SMBNOS), Section of Human Anatomy and Histology, University of Bari “Aldo Moro”, I-70124 Bari, Italy; domenico.ribatti@uniba.it (D.R.); sabrina.lisi@uniba.it (S.L.)

**Keywords:** salivary glands, fibrosis, EMT, Sjögren’s syndrome, autoimmunity

## Abstract

Sjögren’s syndrome (SS) is a systemic chronic autoimmune disorder characterized by lymphoplasmacytic infiltration of salivary glands (SGs) and lacrimal glands, causing glandular damage. The disease shows a combination of dryness symptoms found in the oral cavity, pharynx, larynx, and vagina, representing a systemic disease. Recent advances link chronic inflammation with SG fibrosis, based on a molecular mechanism pointing to the epithelial to mesenchymal transition (EMT). The continued activation of inflammatory-dependent fibrosis is highly detrimental and a common final pathway of numerous disease states. The important question of whether and how fibrosis contributes to SS pathogenesis is currently intensely debated. Here, we collect the recent findings on EMT-dependent fibrosis in SS SGs and explore clinical evidence of multi-organ fibrosis in SS to highlight potential avenues for therapeutic investigation.

## 1. Introduction

Fibrosis is the end result of various chronic autoimmune diseases. Much evidence has been collected demonstrating an abnormal expression of various factors responsible for the activation of fibrotic process in the joints of patients affected by rheumatoid arthritis (RA) [1,2,3,4], in inflammatory bowel disease (IBD), and in conjunction with ulcerative colitis and Crohn’s disease [5,6]. Additionally, renal fibrosis features have often been encountered linked to systemic lupus erythematosus (SLE) nephritis [7,8]. The common denominator in all these fibrotic manifestations in autoimmune diseases appears to be the activation of an epithelial to mesenchymal transition (EMT) process following chronic inflammatory stimulation. The activation of EMT is essential for accurate embryogenesis and tissue repair, and also plays a significant role in the development of fibrosis in mature organs as an outcome of severe chronic disease. This hypothesis was amply demonstrated by experimental animal models, in which the inhibition of EMT is effective in attenuating the progression of tissue fibrosis [9,10]. The concept that chronic injury often triggers EMT cascade, leading to severe organ fibrosis, was recently linked to the atrophy and fibrosis of salivary glands (SGs) [11,12,13], which occurs in the chronic inflammatory autoimmune disease Sjögren’s syndrome (SS) [14].

Based on the scientific evidence that many autoimmune diseases are characterized by secondary fibrotic manifestations in different organs, this review aims to collect scientific evidence of multi-organ fibrotic phenomena in SS due to an excessive production of inflammatory factors. Data reported in the literature seem to support the idea that SS, in addition to being characterized by SG fibrosis, can be associated with fibrosis found in other organs, thus, confirming that SS is a chronic, multisystem autoimmune condition.

## 2. Sjögren’s Syndrome Features

Sjögren’s syndrome (SS) is a systemic chronic autoimmune disorder characterized by the lymphocytic infiltration of the SGs and lacrimal glands that causes glandular damage, leading to xerostomia (dry mouth) and xerophthalmia (dry eyes). Furthermore, SS is also known as “sicca syndrome” or “sicca complex”, because the disease shows a combination of dryness symptoms found in the oral cavity, pharynx, larynx, and vagina. Thus, SS is a systemic disease, involving virtually any organ system. Impaired function is associated with reduced quality of life and symptoms, such as pain, fatigue, and depression, in a comparable way with other diseases, such as SLE or RA [15].

Infectious agents, especially viruses, and genetic and epigenetic factors are supposed to be involved in SS aetiology. The current SS pathogenic model is increasingly known as “autoimmune epithelitis”. This model considers salivary gland epithelial cells as crucial players because they, on the one hand, represent the targets of autoimmune attach and, on the other hand, release various pro-inflammatory factors, exacerbating the immune response [16,17]. Various experimental evidence has demonstrated that overexpression of certain cytokines, such as IFN-gamma and tumor necrosis factor-alpha may contribute to the SG dysfunction observed in SS by disrupting the tight junction structure of epithelial cells [18]. Alterations in the cellular junction integrity lead to significant changes in salivary gland epithelial cells polarity and organization that may affect secretory functionality [19]. This scenario fits well with the inflammatory-related EMT activation program observed in SS, characterized by a loss of epithelial markers, such as E-cadherin and tight junction proteins [20,21,22]. All these phenomena, potentially implicated in the reduction of the normal quality and quantity of saliva in SS, resulted in accelerated development of SG inflammation [18,19].

## 3. Fibrosis and EMT Program Activation

Fibrosis is defined by the accumulation of extracellular matrix (ECM) components, particularly type I collagen and fibronectin by myofibroblasts, at the site of injury [23]. There is a great deal of evidence indicating that myofibroblasts involved in fibrosis are derived from resident epithelial cells that have been transformed through the activation of the EMT program to synthesize ECM factors. The accumulation of fibrotic components can cause malfunction and failure of the organs affected [24,25,26]. Nevertheless, EMT emerges as a decisive factor in activating a pathological fibrotic cascade in chronic inflammatory diseases. Therefore, EMT-dependent fibrosis identifies a condition marked by an uncontrolled and unresolved inflammatory reaction [27]. Furthermore, EMT-dependent fibrosis was found in chronic inflammatory diseases of multiple organs, such as the kidney, liver, lung, intestine, and in SGs [5,6,7,8,9,10,11,12,13,14]. Typically, EMT events occur as part of a repair-associated process in order to rebuild tissues following trauma and inflammatory damage. These events are reparative if the injury is moderate and acute. However, in chronic inflammation, abnormal formation of myofibroblasts provokes a progressive fibrosis that often leads to organ parenchymal destruction and loss of function. On the other hand, inflammation is a potent inducer of EMT and, therefore, inflammation and EMT support each other [9,27].

## 4. Clinical Fibrotic Manifestation in SS

The following paragraphs report the data present in the literature relating to organ fibrosis correlated with SS. The phenomenon has been extensively studied in SGs, where the molecular mechanisms that could trigger fibrosis are now known and have been correlated with EMT. In recent years, cases of secondary fibrosis have also been observed, which could be correlated with the state of chronic inflammation that characterizes SS.

### 4.1. EMT-Dependent Salivary Gland Fibrosis

A clear link between chronic inflammation and fibrosis has been demonstrated in SGs, recently associated with SG atrophy [28,29]. In SS, fibrosis seems to be involved in the decreased secretory function of SGs, which leads to hyposalivation and xerostomia [12]. It is now widely accepted that the development of a fibrotic program in SS is due to the production of fibrogenic mediators by inflammatory and epithelial cells; among these mediators, a prevailing role is played by TGF-β1 [30]. Sisto et al. demonstrated that TGF-β1 promotes salivary gland epithelial cells transition towards mesenchymal cells through the activation of the EMT-dependent fibrosis [31,32,33]. Experiments performed on human salivary gland epithelial cells in vitro demonstrated that TGF-β1 was able to shift salivary gland epithelial cells from the classic cobblestone morphology to a more fibroblast-like morphology characterized by a weakening of cell–cell adhesion. This was supported by the observation that SS SG biopsies show an elevated expression of TGF-β1 [34].

The aberrant upregulation of TGF-β1 in the SS SGs causes EMT via the activation of canonical and non-canonical pathways. As recently demonstrated, the TGF-β1/SMAD/Snail signaling pathway was involved, as confirmed by the detection of a wide distribution of TGF-β1, pSMAD2/3, and SMAD4 proteins in the SS SG tissues. Furthermore, in SS SGs, a strong positivity for EMT-cascade factors and mesenchymal markers was also evidenced, such as SNAIL, vimentin, and collagen type I. Additionally, SS SGs were characterized by a decreased expression of typical epithelial markers, such as E-cadherin [11,35] (Figure 1).

A breakthrough in research has recently been made showing that the loss of epithelial markers and the acquisition of mesenchymal markers was strictly correlated with the grade of SG inflammation. Currently, attempts to explain the development of fibrotic phenomena in SS SGs, induced by the initiation of an EMT program, have focused their attention on the role of several pro-inflammatory cytokines. The results are very encouraging; Sisto et al. demonstrated that IL-17 and IL-22 participate in TGF-β1/EMT-dependent SG fibrosis. Both the cytokines are upregulated in SS and linked to low levels of saliva production; in addition, both IL-17 and IL-22 are abundantly secreted in SS SGs and correlated with the inflammatory degree of the glands [36].

Interestingly, in an experimental model represented by healthy salivary gland epithelial cells in culture, both IL-17 and IL-22 induce morphological changes compatible with those observed in EMT. In particular, using IL-17 as stimulus, in healthy salivary gland epithelial cells, the activation of the canonical TGF-β1/Smad2/3 and non-canonical TGF-β1/Erk1/2 pathway was demonstrated [36]. When testing if other pro-inflammatory cytokines exert their effect on the activation of EMT-dependent fibrosis pathways in SS, interesting results were obtained with IL-6, detected at very high levels in SS SGs. The IL-6 treatment induces a reduced E-cadherin gene transcription and protein synthesis in healthy salivary gland epithelial cells, accompanied by increased levels of vimentin and collagen type I [37] (Figure 1).

### 4.2. Cardiac Fibrosis

Cardiac fibrosis is the accumulation of scar tissue in the heart, and is defined as the imbalance between production and degradation of ECM protein production. Cardiac fibrosis is strongly associated with many cardiac pathophysiologic conditions, and recently, several interesting studies have detected an increased incidence of cardiovascular disease (CVD) morbidity and mortality in patients affected by rheumatic autoimmune diseases, such as SLE and RA [38,39]. In recent years, substantial evidence has emerged demonstrating a link of SS with an increased risk of cardiovascular manifestations, such as stroke and myocardial infarction [40]. Furthermore, intriguing observations have been reported that chronic inflammation in SS patients can trigger a coronary event and, thus, an increased risk of CVD [40], but this needs further investigation [40,41]. Indeed, it was also reported that myocardial injury is typically clinically silent in patients with RA, and this could explain the lack of data on cardiac events in patients with SS, since clinical and pathophysiological characteristics are often shared between RA and SS. In recent papers, it was a high prevalence of myocardial fibrosis in the patients with SS who underwent to cardiac magnetic resonance imaging (cMRI) was observed, which can be used to obtain a quantitative functional evaluation of the myocardium [42,43]. In these studies, emerging data highlight that lymphocytic infiltration into the myocardium is conceivable as a pathological characteristic of myocardial fibrosis in SS patients. The results clearly highlight that the higher the extent of lymphocytic infiltration into salivary glands, the greater the possibility of development of myocardial fibrosis [42]. In fact, myocardial fibrosis is present in patients with SS without cardiac symptoms, and alterations in cMRI data were often linked with SG focus score (FS) ≥ 3 [42,44]. This study suggests a significant association between myocardial fibrosis and the degree of lymphocytic infiltration into the SGs as an important prognosis factor for SS [43]. Yokoe et al., in an interesting study, have obtained several important results from the observation of a representative number of SS patients by the use of non-contrast cMRI, without cardiovascular clinical symptoms [43]. These findings suggest and confirm that cardiac dysfunction and cardiac fibrosis are strongly evident in SS patients. Furthermore, the importance of this study was to demonstrate that myocardial fibrosis could be considered as an extra-glandular event of SS [43], and that cMRI could be a useful tool for detecting asymptomatic myocardial fibrosis in patients with SS with a higher SG FS [42].

### 4.3. Liver Fibrosis

The autoimmune destruction of exocrine glands that occurs in SS often extends to non-exocrine organs. Liver involvement was one of the main extra-glandular events reported in patients with SS [45,46]. In this context, the main causes of liver disease in primary SS are chronic viral hepatitis infections and autoimmune hepatitis [47]. With regards to viral infections, chronic hepatitis C virus infection is often involved in hepatic impairment in SS patients deriving from the Mediterranean area, while chronic hepatitis B virus infection seems to be the main cause of liver involvement in Asian SS patients. Autoimmune hepatitis is the second leading cause of liver damage in SS patients [47]. Liver fibrotic processes depend on the activation of an initial injury of hepatocytes by autoreactive immunological phenomena; these events lead to the proliferation of myofibroblasts and the activation of stellate cells [48]. These manifestations may, in turn, accelerate the deposition of collagen or glycoproteins in the liver, leading to liver fibrosis that interferes with the liver function and contributes to gradual organ failure [49]. The immunological parallel between SS and autoimmune-related hepatitis increases the progression and the development of liver fibrosis in SS. Thus, the assessment of the presence of liver fibrosis and its severity might have a value as prognostic factor in patients with SS. In a recent study, the transient elastography (TE) technique was used, which represents a new non-invasive method for the assessment of hepatic fibrosis in SS patients with normal liver function and structures, and without manifestations of evident liver diseases [46]. Using this approach, a high percentage of SS patients examined present a substantial liver fibrosis, suggesting that the frequency of potential liver fibrosis may have been underestimated in SS patients without clinical symptoms. Furthermore, this study proposed that TE could be used to evaluate the degree of hepatic fibrosis at an earlier stage of SS disease with a notable precision grade [46].

### 4.4. Lung Fibrosis

Pulmonary involvement in SS is an understudied condition with important clinical implications. The common pulmonary manifestations of SS are interstitial lung disease (ILD), airway abnormalities, and lymphoproliferative disorders [50]. Among them, ILD represents a frequent extra-glandular manifestation of SS, with the majority of the studies indicating a prevalence of about 20%, and resulting in significant morbidity and mortality [50,51]. This condition is associated with an injured respiratory function that leads to a poor quality of life and, indeed, is considered a significant cause of fatal outcomes in SS [52]. Therefore, the identification of poor prognostic predictive factors is required in order to provide appropriate management in patients with SS-associated ILD. When ILD includes scar tissue and the injury and damage of the walls of the air sacs of the lung, as well as in the tissue and space around these air sacs, this condition is known as pulmonary fibrosis. Pulmonary fibrosis is part of this wide group of more than 200 ILD. Efforts have been made to characterize the relationship between SS and ILD, with an emphasis upon idiopathic pulmonary fibrosis (IPF). Roca et al. highlighted that ILD is observed in a significant percentage of SS patients, and that this condition is associated with severe lung injury that develops versus fibrosis pulmonary [53]. Recently, an interesting study was addressed to systematically evaluate the incidence and characterize ILD fibrosis phenotype in a well-defined SS-ILD cohort [54]. These data have revealed that pulmonary disease is commonly linked with SS, resulting in a wide variety of clinical manifestations [54]. Firstly, symptomatic lung involvement triggers scar tissue and injury, provoking an evolution toward a progressive fibrosing phenotype in the lung identified in 13% of SS patients and so confirming previous investigations [55,56]. The second important implication is the need for effective SS screening in patients presenting apparently idiopathic ILD [54]. Subsequently, recent studies from different countries have, however, all observed that the prognosis of pulmonary involvement is not favorable in patients with SS [57]. Thus, early ILD and IPF detection is very important in SS disease evolution [58]. However, it remains controversial whether all SS patients should undergo a systematic search for lung involvement [59] with the view to redefine disease recognition strategies.

### 4.5. Kidney Fibrosis

Renal involvement is an extra-glandular condition well recognized in SS patients. The most common histopathological condition is an interstitial lymphocytic infiltrate with tubular atrophy and, consequently, renal fibrosis that leads to a slow progressive deterioration in kidney function [60].

Kidney disease typically manifests 2–3 years after the beginning of the involvement of the exocrine glands, and slowly leads to decreased renal function. Kidney disease occurs in 5% of patients with SS, with a broad range of clinical conditions [60]. The most frequent event of nephropathy in SS is tubule interstitial nephritis (TIN), characterized by lymphoplasmacytic infiltration of the kidney showing similarity to the lymphoid infiltration that occurs in the SGs. Patients with SS associated with TIN have significant renal fibrosis, and, as a consequence, show organ impairment and lymphocytic infiltration leading to acute or chronic forms of TIN [61,62,63]. New emergent observations suggest that infiltration in the renal tubules is mostly caused by CD4+ T lymphocytes, features of the pathophysiological process in SGs [61,64]. Unfortunately, TIN remains a condition often undiagnosed due to its inauspicious clinical course [60,62]. Recently, a wide Taiwanese cohort study indicated that patients with SS are more likely to develop chronic kidney disease as a consequence of TIN, and found that a progressive decline in kidney function occurred in 15% of SS patients [62].

A schematic representation of all the identified secondary fibrosis in SS is shown in Figure 2.

## 5. Conclusions

Sjögren’s syndrome is a chronic inflammatory autoimmune disease of variable severity and course. Although SS continues to be a challenging disease, there is now a better knowledge of its causes, earlier recognition of its symptoms, and more effective therapeutic treatments. In this review update, we are discussing evolving concepts in SS which is considered, in fact, to be a systemic disease with a fibrotic evolution of SGs. We discuss more recent studies, mostly published within the last 5 years, highlighting the possibility that secondary organ fibrosis could be a feature of SS. The clinical implication of this review article is, therefore, to summarize the current state of knowledge of molecular mechanisms involved in SG fibrosis in SS. The relationship between inflammation, EMT, and fibrosis has been established in several autoimmune diseases. The majority of studies on EMT-dependent fibrosis in SS have been carried out in the SGs, and, actually, the possibility that the same pathways operate during other secondary fibrotic processes in SS, including cardiac, pulmonary, renal, or hepatic fibrosis, has not yet been investigated. The purpose of our review is to draw precise attention to a possible and probable involvement of an EMT program in the fibrotic evolution of secondary diseases associated with SS, and to emphasize that a multidisciplinary approach is needed to identify the secondary fibrotic forms observed in SS disease. These data will help physicians better understand the disease, and to identify novel therapeutic protocols to block fibrosis in SS patients.

## Figures and Tables

**Figure 1 jcm-11-03551-f001:**
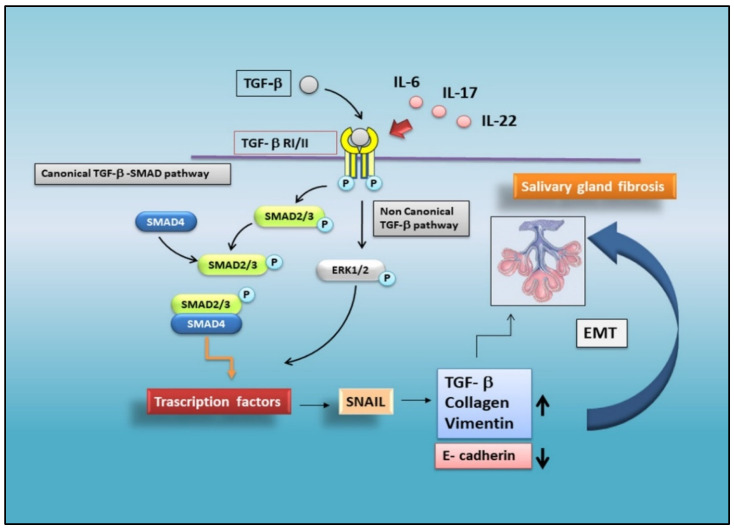
**Schematic representation of TGF-β-mediated EMT signalling in SS.** In a situation of chronic inflammation, TGF-β activates the canonical SMAD2/3 and the non-canonical ERK-mediated pathways, triggering the EMT process in salivary gland epithelial cells. The activation of transcription factors (such as SNAIL), promotes the prolonged induction of EMT, repressing epithelial marker genes and activating genes linked to the mesenchymal phenotype. Pro-inflammatory cytokines, such as IL-17, IL-22, and IL-6, induce EMT-dependent severe fibrosis in SGs.

**Figure 2 jcm-11-03551-f002:**
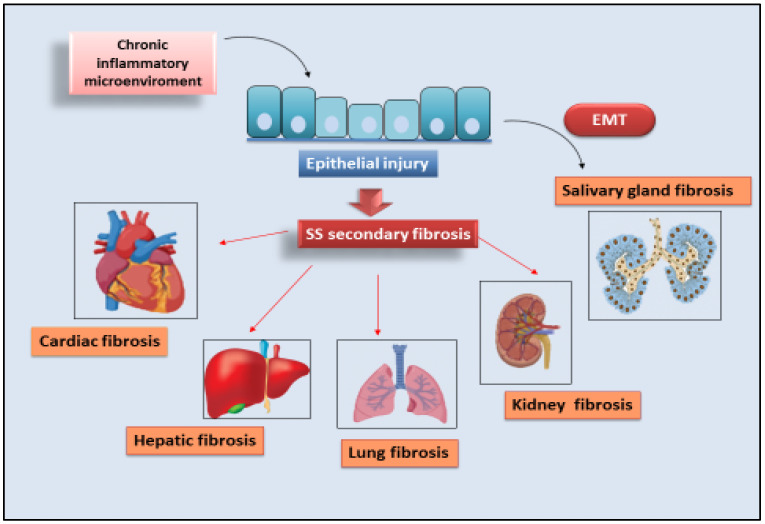
**Secondary organ fibrogenesis in SS**. Chronic inflammatory microenvironment cooperates for the progression of organ fibrosis in SS patients. Injury events lead to organ damage, inflammation, and fibrosis in the liver, kidney, lung, heart, and SGs.

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
