# Peer review of "Sjögren’s Syndrome-Related Organs Fibrosis: Hypotheses and Realities"

_jcm, 2022, doi:10.3390/jcm11123551_

Round 1

Reviewer 1 Report

I read with interest the article by Sisto M et al. They emphasized that multiple organs fibrosis in Sjogren's syndrome, besides salivary glands, and highlighted potential avenues of therapeutic investigation.The authors reviewed the mechanism of fibrosis, but need to add some new insights. Regarding fibrosis in internal organ, the authors just showed indirected evidence detected by imaging modality, but not pathological data. They didn't mention how to deal with the early fibrosis in this scenario. 

Author Response

I would point out that the association between chronic inflammation and organ fibrosis has attracted great attention in recent years, and many studies have shown that fibrosis secondary to a chronic inflammation, that characterizes most autoimmune diseases, occurs through the activation of an EMT program. In SS, specific studies have been carried out exclusively on salivary glands and this represents a pioneering study carried out by my research group. This is due to the difficulty to find pathological data which correlate liver, cardiac, lung or renal disfunction with organ fibrosis in SS patients; in addition, we contacted a large pathological anatomy laboratory asking to evaluate if, in the archive, there were cases of SS patients undergoing biopsies in secondary organs such as liver, heart, lungs or kidney that allowed us to evaluate the molecular mechanisms underlying secondary organ fibrosis but without success. The data in the literature concern either post-mortem cases or diagnostic investigations that evaluated secondary fibrosis in SS patients due to liver or heart or kidney or lung damage. The purpose of our review is precisely to encourage other research groups not to underestimate fibrotic manifestations secondary to SS in the hope of clarifying the molecular mechanisms underlying these fibrotic evolutions.

Reviewer 2 Report

the article is well written

manuscript content looks apt in understanding the hypothesis towards the development of fibrosis in various vital organs in patients with ss, but I raise an issue towards self-citation about similar work from the same author, and content requires grammatical corrections. overall recommendation rest upon the editor to include or not to include article with self citations

Author Response

I thank the reviewer for the positive comments regarding our manuscript. We have carried out a grammar check throughout the manuscript, and we hope that it will be sufficient to eliminate the errors found. I would point out that the association between chronic inflammation and organ fibrosis has attracted great attention in recent years, and many studies have shown that fibrosis secondary to a chronic inflammation situation that characterizes most autoimmune diseases occurs through the activation of an EMT program. In SS, specific studies have been carried out exclusively on salivary glands and this represents a pioneering study carried out by my research group. This is due to the difficulty in finding liver, cardiac or renal biopsies of SS patients that allow us to assess whether secondary fibrosis is actually due to an EMT process. The data in the literature concern either post-mortem cases or diagnostic investigations that evaluated secondary fibrosis in SS patients due to liver or heart or kidney damage. The purpose of our review is precisely to draw attention to a possible and probable involvement of an EMT program in the fibrotic evolution of secondary diseases associated with SS, in the hope that other research groups will begin to investigate the molecular mechanisms underlying this process, as my research group is doing on the salivary glands. I have made this clearer in the conclusion.

Reviewer 3 Report

This manuscript is a well-written review report demonstrating fibrosis in Sjӧgren's syndrome (SS) that causes several clinical manifestations in systemic organs. Authors concisely and clearly introduce the outline of fibrosis due to SS and its molecular mechanisms including the TGF-β1/SMAD/Snail signaling pathway and the roles of several cytokines. In addition, extra-glandular manifestation of SS was also documented, which might affect prognosis of patients with SS.

Queries and recommendations

1.        The use of uncommon abbreviations such as “SGEC” should be avoided because it makes readers irritate.

2.        I recommend you to move the sentence on the line 113-115 to the next paragraph.

3.        On the 176 and 177, authors described the relationships between SS and viral and autoimmune hepatitis.  However, there is no description of the relationships between SS and viral hepatitis.

4.        Please comment whether EMT, which is a very interesting phenomenon, relates secondary fibrosis in liver, lung and kidney where epithelium is injured by inflammatory irritation.

Author Response

Queries and recommendations

  1. The use of uncommon abbreviations such as “SGEC” should be avoided because it makes readers irritate.

As suggested, we have eliminated the SGEC abbreviation by replacing it with the full “salivary gland epithelial cells”

  1. I recommend you to move the sentence on the line 113-115 to the next paragraph.

Done.

  1. On the 176 and 177, authors described the relationships between SS and viral and autoimmune hepatitis. However, there is no description of the relationships between SS and viral hepatitis.

I have reported in the text a brief description of the causes of viral hepatitis in SS patients and replaced the reference 47 with a more specific one.

  1. Please comment whether EMT, which is a very interesting phenomenon, relates secondary fibrosis in liver, lung and kidney where epithelium is injured by inflammatory irritation.

I would like to thank the reviewer for his interesting observation and point out that the association between chronic inflammation and organ fibrosis has attracted great attention in recent years and many studies have shown that fibrosis secondary to a chronic inflammation situation that characterizes most autoimmune diseases occurs through the activation of an EMT program. In SS, specific studies have been carried out exclusively on salivary glands and this represents a pioneering study carried out by my research group. It is currently difficult to find liver, cardiac or renal biopsies of SS patients that allow us to assess whether secondary fibrosis is actually due to an EMT process. The data in the literature concern either post-mortem cases or diagnostic investigations that evaluated secondary fibrosis in SS patients due to liver or heart or kidney damage. The purpose of our review is precisely to draw attention to a possible and probable involvement of an EMT program in the fibrotic evolution of secondary diseases associated with SS, in the hope that other research groups will begin to investigate the molecular mechanisms underlying this. process, as my research group is doing on the salivary glands. I have made this clearer in the conclusion.

Round 2

Reviewer 1 Report

I am satisfied with the author's reply. I will suggest to accept the article.